# Endemic and Threatened *Amazona* Parrots of the Atlantic Forest: An Overview of Their Geographic Range and Population Size

**Viviane Zulian** [1,*] , **David A. W. Miller** [2] and **Gonçalo Ferraz** [1]

1 Programa de Pós-Graduação em Ecologia, Instituto de Biociências, Universidade Federal do Rio Grande do Sul, CP 15007, Porto Alegre 91501-970, RS, Brazil; goncalo.ferraz@ufrgs.br
2 Department of Ecosystem Science and Management, Pennsylvania State University, 411 Forest Resources Building, University Park, State College, PA 16802, USA; dxm84@psu.edu
* Correspondence: zulian.vi@gmail.com

**Abstract:** *Amazona* is the largest genus of the Psittacidae, one of the most threatened bird families. Here, we study four species of *Amazona* (*Amazona brasiliensis*, *A. pretrei*, *A. vinacea*, and *A. rhodocorytha*) that are dependent on a highly vulnerable biome: the Brazilian Atlantic Forest. To examine their distribution and abundance, we compile abundance estimates and counts, and develop site-occupancy models of their geographic range. These models integrate data from formal research and citizen science platforms to estimate probabilistic maps of the species' occurrence throughout their range. Estimated range areas varied from 15,000 km² for *A. brasiliensis* to more than 400,000 km² for *A. vinacea*. While *A. vinacea* is the only species with a statistical estimate of abundance (~8000 individuals), *A. pretrei* has the longest time series of roost counts, and *A. rhodocorytha* has the least information about population size. The highest number of individuals counted in one year was for *A. pretrei* (~20,000), followed by *A. brasiliensis* (~9000). Continued modeling of research and citizen science data, matched with collaborative designed surveys that count parrots at their non-breeding roosts, are essential for an appropriate assessment of the species' status, as well as for examining the outcome of conservation actions.

**Keywords:** *Amazona*; Psittacidae; species distribution models; data integration models; occupancy models; citizen-science; population size; count data

## 1. Introduction

Three hundred and ninety-five species of parrots, macaws, and parakeets constitute the Psittacidae family, the largest non-passerine bird family in the world [1]. With 27% (108) of its species threatened with extinction [1], the Psittacidae is the bird family with the highest absolute number of threatened species, that is, species classified as 'vulnerable', 'endangered', 'critically endangered', or 'extinct in the wild', by the International Union for Conservation of Nature and Natural Resources (IUCN). In proportional terms, the Psittacidae come only after the much smaller families of albatrosses and cranes with, respectively, 68% and 66% of their species threatened. Habitat loss and nest poaching are two key factors endangering Psittacidae populations [2,3]. Being dependent on forest habitats, most Psittacidae species require natural cavities to nest [3] and are thus directly impacted by forest clearance [2] and selective logging [4], caused primarily by agro-industrial expansion [5,6]. Nest poaching disproportionately affects species that are colorful, with large body size, relative ease of capture, and that sell for the highest prices [7,8].

The most diverse genus among the Psittacidae is the neotropical genus *Amazona*, or Amazon parrots, with 36 species distributed from northern Argentina to northern Mexico [1]. One half (18) of the *Amazona* species are globally threatened, and 25 species have decreasing population sizes, according to the IUCN Red List [1]. Nest poaching

has been reported by Wright et al. [8] as the main cause of mortality in four species: *A. vinacea*, *A. kawalli*, *A. ochrocephala,* and *A. auropalliata*. Habitat loss is also a threat to the genus, especially in those biomes that have been more subjected to deforestation, such as the Atlantic Forest of Brazil. Home to seven *Amazona* species [9], the Atlantic Forest is the second largest rainforest in South America [10,11] and is a global biodiversity hotspot [12]. The biome has lost almost 90% of its forest cover since the onset of European colonization [12], and only 1% of its original extent is presently included in protected areas [10]. According to one projection to 2070 [13], the Atlantic Forest region will lose bird habitat at the rate of 1.2% to 3.3% per decade—the highest rate of loss estimated by that study for any region of the world. Realizing the potential impact of land use in the Atlantic Forest on parrot populations [14], as well as the relative importance of the genus *Amazona* among the Psittacidae, we direct our attention here to what we consider to be the most emblematic *Amazona* species of the Atlantic Forest biome: *A. brasiliensis*, *A. rhodocorytha*, *A. vinacea*, and *A. pretrei*. They are endemic to the Atlantic Forest [15] and classified by the IUCN, respectively, as Near-Threatened, Vulnerable, Endangered, and Vulnerable.

Geographic range and population size are two key descriptors of the state of any living species. Since their temporal trajectories offer evidence of population trends, these two variables inform four out of the five criteria used by the IUCN in assigning species to threat categories [16]. Notwithstanding, the IUCN Red List profiles of these four species in this study reveal substantial uncertainty about their geographic ranges and limited information about how the estimated population sizes were obtained. Our goal here is to fill this knowledge gap to the extent that is possible by compiling information from the ornithological literature and citizen-science platforms. We review information on population sizes based on published abundance estimates and counts of all species. To address geographic ranges, we draw new maps for the four species. Our maps express the species' distribution as occupancy probability per municipality. The statistical models used for producing the new maps integrate data from three different citizen-science platforms (eBird, Wikiaves, and Xeno-Canto) as well as from formal research databases, where available. We hope that improved knowledge about abundance and distribution of *Amazona* species in the Atlantic Forest will help direct future monitoring and conservation efforts, as well as strengthen the basis for threat assessments.

## 2. Materials and Methods

### 2.1. Study Area and Data Collection

We organized information about the population size and the geographic range of *Amazona brasiliensis*, *A. pretrei*, *A. rhodocorytha*, and *A. vinacea* following two different approaches. For population size, we compiled all the information about counts or abundance estimates that we could find for each species, including results from peer-reviewed papers, reports, books, and academic theses (Supplementary Tables S1–S3). Count data were collected by four different research teams, during scientific research or monitoring programs. The counts were performed at regularly used roosts or near points of frequent flyover by parrots, at dawn and dusk. For geographic range mapping, we compiled detection–non-detection data from citizen-science platforms and research project databases. Such data were analyzed separately for each species, with a site-occupancy, data-integration model following Zulian, Miller, and Ferraz [17]. We varied the geographic extent, or focal area, used to fit each species' model (Figure 1). Focal areas included either all the states or provinces where the species were detected (*A. rhodocorytha* and *A. vinacea*) or all the municipalities within 150 km of the closest detection (*A. brasiliensis* and *A. pretrei*). These areas ranged from a little over 160,000 km$^2$, for *A. brasiliensis*, to more than 1.5 million km$^2$, in the case of *A. vinacea* (Figure 1). We are confident that the extent for each species covers the entirety of each species' potential area of occurrence. The *A. vinacea* range map that we present here is the only map in this paper that combines formal research and citizen-science data. This map is identical to that shown by Zulian, Miller, and Ferraz [17], in a study focused on devising optimal methods for fitting distribution models to multiple

data streams, which informs the approach that we used here. Geographic range analyses for the other three species are based uniquely on citizen-science data, as explained below.

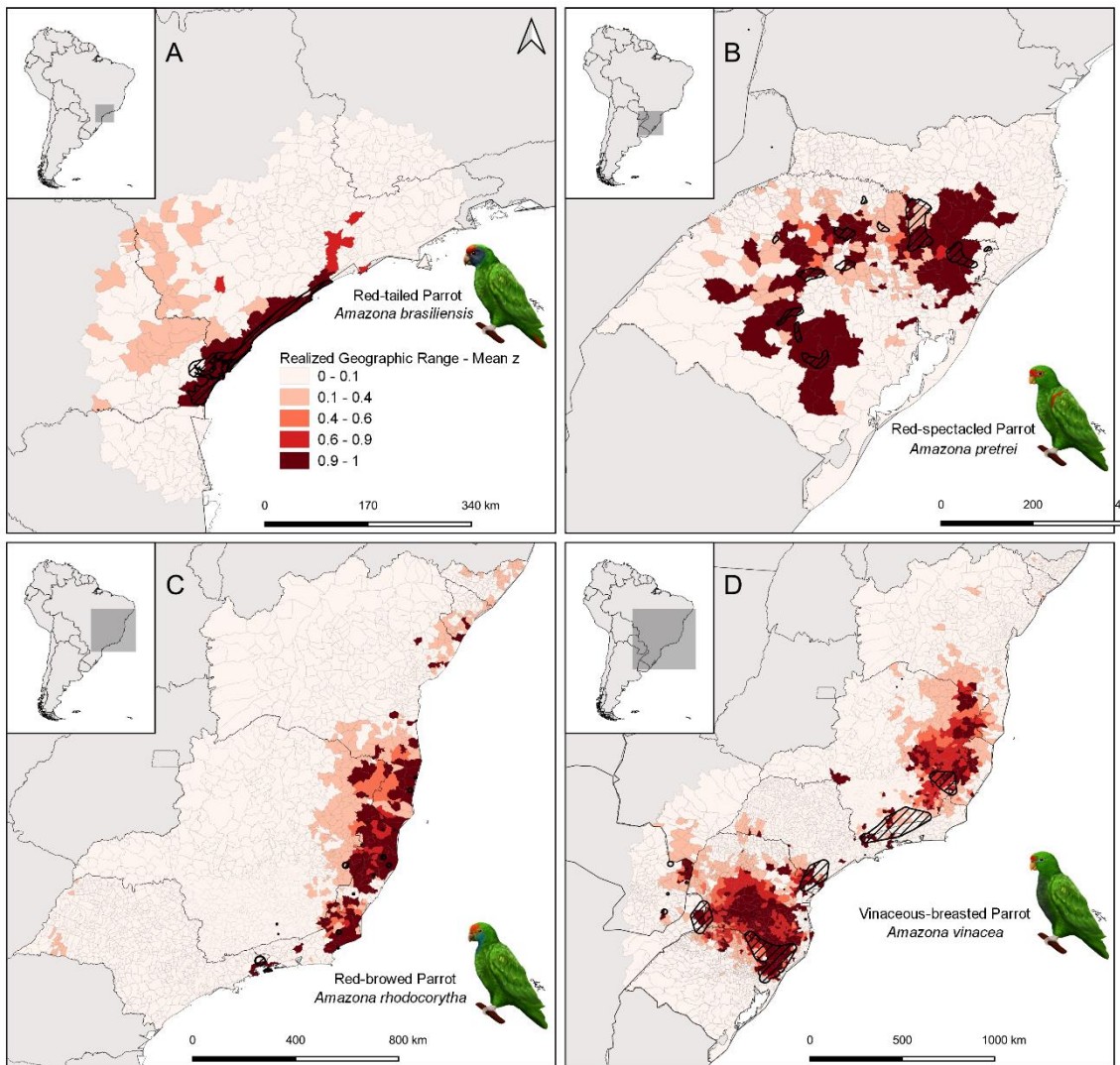

**Figure 1.** Geographic ranges of the four study species represented by the mean of the true occupancy state (*z*) estimated for each municipality. Intermediate values—of $z \sim 0.5$—indicate the highest uncertainty about occupancy by each species. Black dashed polygons are the Extant range of each species according to the IUCN Red List of Threatened Species [1].

We obtained records of *A. vinacea*, *A. brasiliensis*, *A. rhodocorytha*, and *A. pretrei* from citizen-science platforms eBird [18], WikiAves [19], and Xeno-canto [20], corresponding to the period between 1 January 2008 and 31 December 2018. For *A. vinacea*, we also included a formal research dataset derived from roost counts and described by Zulian et al. [21]. Our sampling unit is the municipality, where the number varied from 3701 to 405 depending on the species. Citizen-science platforms store data resulting from field visits with highly variable duration, distance covered, observation technique, and observer experience. This lack of standardization requires platform-specific data processing solutions. In particular, eBird data come in the form of checklists, which contain information about observation effort per list. The number of lists per municipality varied from 1 to 3245, with a mean of 33 lists, collected at different times of the year by different observers. Wiki-Aves and Xeno-canto, on the other hand, gather records for a municipality in the form of individual species observations that are not aggregated in any form of observation 'session' per municipality and observer. As a result, we have the equivalent of replicate visits for eBird, but not for the other two platforms, where each municipality has only one 'visit'.

Data processing consisted of some filtering, formatting data matrices, and obtaining effort covariates for all platforms. Starting with eBird, we excluded incomplete checklists, checklists without location information, and checklists that potentially spanned more than one municipality due to long distance (>12 km) or long time traveled (>360 min). We set up a matrix of detection–non-detection histories based on eBird data for each parrot species. In this matrix, municipalities appeared in rows and consecutive checklists of each municipality in columns. Matrix elements were '1', for municipalities and checklists where the parrot species were detected, or '0' where not detected. We calculated three covariates of sampling effort for each eBird checklist and municipality: the total number of species recorded, the number of minutes spent observing, and the number of kilometers traveled. For WikiAves and Xeno-canto, data filtering consisted of deleting sightings of individuals reported as escaped from captivity. Since WikiAves receives photographic and audio records of species, we organized data into two vectors per parrot species, one for the number of photographs and one for the number of audio recordings of that species in each unique municipality. For the WikiAves data, we calculated two covariates of effort: the number of photos and the number of audio recordings of all species, per municipality. Finally, the Xeno-canto platform hosts only audio recordings of bird sounds, so its detection data were easily organized into a single vector per parrot species, holding that species' number of audio recordings per municipality. We also collected the total number of recordings of any species uploaded for each municipality for use as a covariate of Xeno-canto sampling effort. For the *A. vinacea* research data, we created a detection–non-detection matrix with municipalities as rows and counts as columns. Matrix cells corresponding to counts with at least one parrot received a detection ('1'), and those with no parrots received a non-detection ('0'). Here, we used the count's duration, in minutes, as a covariate of sampling effort (see Zulian, Miller, and Ferraz [17] for details).

### 2.2. Data Analysis

We drew range maps representing the estimated probability of site (or municipality) occupancy by each species during the eleven-year study period. We follow a static approach as originally described by MacKenzie et al. [22] and define 'occupancy' as the probability that a site was occupied by the given species at any point during the whole eleven-year study period. One of the species—*A. pretrei*—is known for its within-year shifts in distribution, which result in exceptionally large concentrations of individuals during the non-breeding season. Therefore, for this species alone, we estimated both the full-year distribution for the species and seasonal range maps. Seasonal distributions were obtained with the same modeling approach applied to four non-overlapping temporal subsets of the data, each corresponding to one trimester of the year and including information from all years. At the core of our statistical approach to site occupancy, there is a process model of the true occupancy state, $z_i$, of each municipality, $i$, which takes the value of '1' for those municipalities that are occupied by the species of interest, and '0' for those that are not. This state follows a Bernoulli distribution with a mean $\psi_i$:

$$z_i \sim Bernoulli\,(\psi_i). \tag{1}$$

The occupancy probability in each municipality $i$, $\psi_i$, varies according to $n$ environmental covariates, $X_{n,i}$, according to a generalized linear model with a logit link function. Since the four species of parrots are associated with Atlantic Forest and altitude [23–25], we included the Atlantic Forest cover and average altitude as covariates of municipality $i$ occupancy. We also included the Araucaria Forest cover as a covariate of occupancy by *A. pretrei* and *A. vinacea*, since they rely heavily on Araucaria seeds for food during the winter [23,26,27], and a Dense Forest cover as a covariate of occupancy by *A. brasiliensis*, because this species is apparently associated with dense, lowland coastal forest [25,28,29]. We obtained Atlantic Forest cover data from Ribeiro et al. (in prep.), and Dense Forest cover data from the Brazilian Instituto Brasileiro de Geografia e Estatística (https://www.ibge.gov.br/ (accessed on 30 June 2021)) [30]. Average municipality altitude, $x$, in meters, is from DIVA-

GIS (https://www.diva-gis.org/ (accessed on 18 November 2019)) [31], log-transformed as $log(x + 1)$. Our linear model of occupancy also included a spatial random effect to account for unexplained spatial autocorrelated variation ($\delta_i$):

$$logit(\psi_i) = \beta_0 + \beta_1 * X_{1,i} + \beta_2 * X_{2,i} + \ldots + \delta_i. \tag{2}$$

In this model, occupancy covariates measured at municipality $i$ are given by $X_{1,i}$, $X_{2,i}$ ..., and $\beta_0$, $\beta_1$, $\beta_2$ ... are estimated coefficients. The spatial component of our model follows a conditional auto-regressive (CAR) distribution [17] and was used to estimate correlated spatial variation in the data that is not explained by our covariates. To avoid confounding effects of municipality size variability and to gain replication within spatial units in the CAR analysis, we represented the spatial random effect using a hexagonal lattice overlaid on the study area, with municipalities assigned to the lattice cell that matches their centroid. Hexagonal cells measured 0.5° latitude across, and all the first-order neighbors of each cell were given a weight of 1 when fitting the CAR model.

Ours is a data-integration approach because it models detections from different databases with a joint-likelihood that shares the same occupancy process described above [32,33], for each parrot species. Within each database, detection was expressed as a conditional probability, $p_j^*$, of detecting the species as a function of an estimated amount of sampling effort, $E_j$, for visit $j$ [17,33,34]:

$$p_j^* = 1 - (1 - p)^{Ej}, \tag{3}$$

where $p$ is the probability of detection per unit of effort. Since we are using indirect, and sometimes several metrics of effort for each data source (our effort covariates), we estimated the parameter $E_j$ for each sample $j$ as a linear function of the covariates. Thus, for each dataset, $DS_n$, with $n$ varying between one and four (roost counts, eBird, WikiAves, and Xeno-canto), we have:

$$E_j^{DS_n} = \alpha_1 * X1_j + \alpha_2 * X2_j + \alpha_3 * X3_j, \tag{4}$$

were $X1_j$, $X2_j$, and $X3_j$ are effort covariates measured on visit $j$. We used one to three covariates depending on data type. We fixed $p$ at a value of 0.5, so that the $\alpha_1\alpha_3$ coefficients express the relationship between covariates and the effort necessary to reach a detection probability of 0.5 per unit of effort. Without fixing $p$, Equation (3) becomes over-parameterized. Having modeled a conditional probability of detection, $p_j^*$, we can represent the detection–non-detection data, $Y_{ij}$, as the outcome of a Bernoulli distribution, that accounts for the true state of each municipality, $z_i$, and the conditional probability of detection, as follows:

$$Y_{ij} \sim Bernoulli\left(z_i \times p_j^*\right). \tag{5}$$

We fitted all the models using a Bayesian estimator coded in the BUGS language and implemented on WinBugs software [35]. Inference was based on draws from the posterior distribution of model parameters using a Markov Chain Monte Carlo (MCMC) algorithm with three chains, 200,000 iterations, and a burn-in phase of 150,000 (see code in Supplementary S1 in the Supplementary Materials). All results presented here correspond to chains that converged to an R-hat lower than 1.1. We draw maps of 'realized occupancy' given by the mean of the estimated $z_i$ for each municipality and estimated the area of each species' geographic range as the sum of all municipality areas weighted by each municipality's predicted occupancy, $\psi_i$, estimate.

## 3. Results

We used a total of 100,289 samples, collected across 3701 municipalities, to inform the estimation of geographic ranges of the four parrot species that we studied. The datasets showed a wide coverage, with more than 90% of the municipalities in each species' study area having at least one sample (Table 1). Alone, the *A. vinacea* dataset accounted for

almost 50% of the samples and 58% of the detections, while *A. pretrei* had the smallest dataset, with 18% of the samples and 10% of the detections (Table 1). *A. brasiliensis* had the third highest number of samples, but only 15 municipalities with at least 1 detection. *A. rhodocorytha* had the second smallest sample size, but the second largest number of detections (Table 1).

Our estimated geographic ranges differed from the Extant area calculated from the range maps reported by the IUCN for all four species (Table 1). *A. vinacea* had the largest estimated range, encompassing more than 400,000 square kilometers [17], followed by *A. rhodocorytha*, with approximately 134,000 square kilometers (Figure 1). The discrepancies between the IUCN Extant area and our estimates are not negligible: while our geographic range estimate is three times larger than the IUCN value for *A. vinacea*, it is six times larger for *A. pretrei*. The biggest discrepancy is for *A. rhodocorytha*, for whom the IUCN reports a range 50 times smaller than our estimate. Geographic ranges are an outcome of history and environmental constraints. Our results show how the environmental covariates of Atlantic Forest cover, Araucaria Forest cover, and Altitude help explain the distribution of *A. vinacea*, with all three having strong and positive effects on site-occupancy probability (Table 2). Based on our models, species' detection data, and environmental covariate information, there is no evidence of other statistically distinguishable effects of environmental factors on site occupancy by any of the four species of parrots (i.e., the 95% credible intervals of other coefficients in Table 2 are nearly centered on zero).

**Table 1.** Sample size, spatial coverage, and number of detections for the four parrot species. Sample size is the number of samples collected form the citizen-science and research datasets, as defined in the text. Coverage is the proportion of municipalities in each study area with at least one sample. The labels $n_{det}$ and $n_{muni}$ show, respectively, the number of parrot detections and the number of municipalities with at least one detection. The last two columns show geographic ranges sizes: the IUCN Extant area is given in each species' online entry to the IUCN Red List of Threatened Species. The estimated geographic range is the sum of municipality areas weighted by the estimated probability the species occurred in each municipality (given here by the mean $\pm$ standard deviation of the a posteriori distribution of range size, followed by its 95% credible interval (in parentheses)).

| Species | Sample Size | Coverage % | $n_{det}$ | $n_{muni}$ | IUCN Extant Area (km$^2$) | Estimated Geographic Range (km$^2$) |
|---|---|---|---|---|---|---|
| *Amazona brasiliensis* (Red-tailed Parrot) | 16,705 | 99.7 | 192 | 15 | 4750 | 15,627 $\pm$ 8843 (3127–31,414) |
| *Amazona pretrei* (Red-spectacled Parrot) | 5477 | 92.7 | 187 | 73 | 10,430 | 66,203 $\pm$ 11,425 (45,727–90,367) |
| *Amazona rhodocorytha* (Red-browed Parrot) | 30,867 | 94.2 | 346 | 86 | 2672 | 134,355 $\pm$ 13,922 (109,288–162,828) |
| *Amazona vinacea* (Vinaceous-breasted Parrot) | 47,240 | 91.9 | 1007 | 339 | 145,700 | 434,670 $\pm$ 28,911 (382,887–496,550) |

**Table 2.** Coefficients of the occupancy equation in each species model. The numbers show the mean $\pm$ standard deviation and 95% credible intervals (in parentheses) of the a posteriori distribution of each parameter.

| Species | Atlantic Forest | Dense Forest | Araucaria Forest | Altitude |
|---|---|---|---|---|
| *Amazona brasiliensis* | $-0.63 \pm 2.23$ (−5.39–3.48) | $-1.33 \pm 3.14$ (−7.71–4.77) | — | $0.23 \pm 0.44$ (−0.72–0.94) |
| *Amazona pretrei* | $-0.55 \pm 1.08$ (−2.70–1.56) | — | $0.47 \pm 1.05$ (−1.55–2.53) | $0.15 \pm 0.18$ (−0.21–0.52) |
| *Amazona rhodocorytha* | $0.84 \pm 0.91$ (−1.70–1.84) | — | — | $-0.14 \pm 0.20$ (−0.51–0.25) |
| *Amazona vinacea* | $2.11 \pm 0.86$ (0.37–3.79) | — | $2.13 \pm 0.98$ (0.29–4.10) | $0.85 \pm 0.12$ (0.58–1.05) |

The subdivision of *A. pretrei* data into trimesters generates four substantially different geographic range maps (Figure 2). During the early breeding season months of July to September, the species is at its most dispersed (Figure 2A). During this period, 39 municipalities throughout the focal area have realized occupancy greater than 0.9 (i.e., mean z > 0.9), even though almost all of them are in the state of Rio Grande do Sul. During the Fall months of April to June, however, *A pretrei* individuals appear aggregated in only 12 municipalities that have realized occupancy greater than 0.9 (Figure 2D). These municipalities form four disjunct clusters in the Rio Grande do Sul and Santa Catarina states. The transition from the aggregated to the dispersed state is faster than the transition from dispersed to aggregated, which takes place from October to March and is represented by the intermediate ranges in panels B and C, of Figure 2.

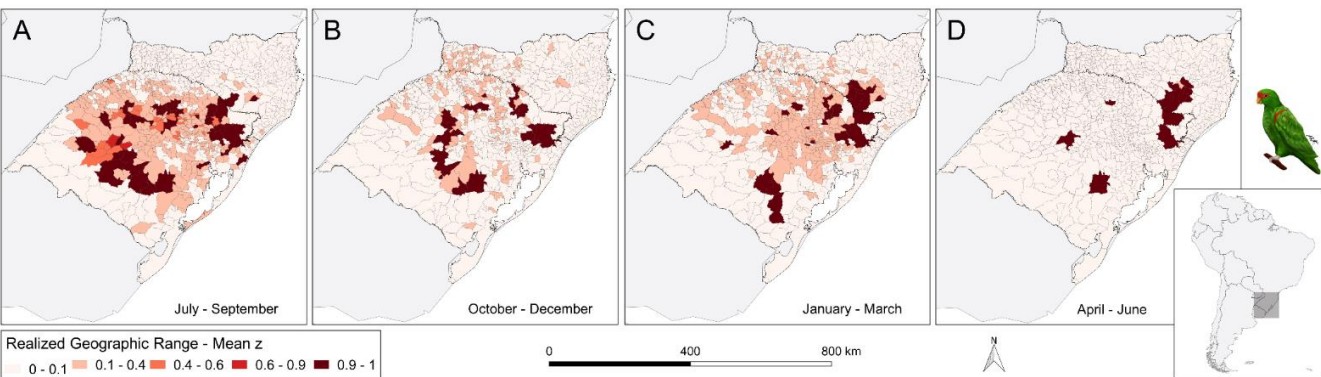

**Figure 2.** Seasonal variation of the geographic range of the Red-Spectacled Parrot (*Amazona pretrei*) as shown by the mean of the true occupancy state (z) estimated for each municipality. Each panel represents a trimester, the sequence starting with July–September (**A**), when the species is most dispersed, and proceeds in three-month intervals to October-December (**B**), January-March (**C**) until April–June (**D**), when it aggregates in only a few municipalities. Darker tones of red indicate higher mean z; intermediate tones indicate the highest uncertainty about species presence.

There is much less information about the abundance of Atlantic Forest *Amazona* species than about their geographic range. The species for which we could assemble the longest time series of roost counts was *A. pretrei*, which has a long-term monitoring program led by the same team of researchers since the mid-1990s (Figure 3B). *A. pretrei* is also the species with highest counted number of individuals. Its earliest counts, performed in 1971 by Forshaw and Cooper (ref [36] cited by [37]), returned between 10,000 and 30,000 individuals (Figure 3B). Later, during the 1970s and 1980s, Belton [38] and Varty et al. [37] reported a decline in the number of individuals counted, with recovery during the 1990s. Since 1997, the yearly sum of *A. pretrei* counts has varied around 20,000 individuals (Figure 3B) [39–42]. The second longest time series of roost counts is that of *A. brasiliensis*. This species also has an ongoing monitoring program, coordinated by the same team throughout the last two decades [43]. The sum of *A. brasiliensis* counts has varied, always below 10,000 individuals, over the last three decades [28,29,43–61]. Figure 3A shows a tendency towards increasing counts, but one should not rush to interpret this as evidence of population growth because the count reports do not incorporate corrections for variation in effort through time. *A. vinacea* has the shortest time series of roost counts [21,27,42,62–64] (Figure 3C) but is the only species with a published statistical estimate of population size, which does account for temporal changes in sampling effort, as well as for detection errors [21]. There are two estimates, for the non-breeding seasons of 2016 and 2017, both in the vicinity of 8500 individuals and with 95% credible intervals entirely below the 10,000-individual mark.

We could not assemble a time series of *A. rhodocorytha* counts, as the few published count results were obtained in sparse locations that were not revisited in different years. In 1998, Marsden et al. [65] searched for the species in two separate sites covering 427 km$^2$ of Bahia and Espírito Santo states, reporting distance-sampling estimates of, respectively, 238 $\pm$ 174 and 5990 $\pm$ 1680 (mean $\pm$ standard error) individuals. Later,

in 2008, Klemann-Júnior et al. [66] counted 2295 individuals for all of Espírito Santo state. The *Plano de Ação Nacional para a Conservação dos Papagaios da Mata Atlântica* considers that the *A. rhodocorytha* population size is around 10,000 individuals, based on expert opinion [41], but no more demographic information is available.

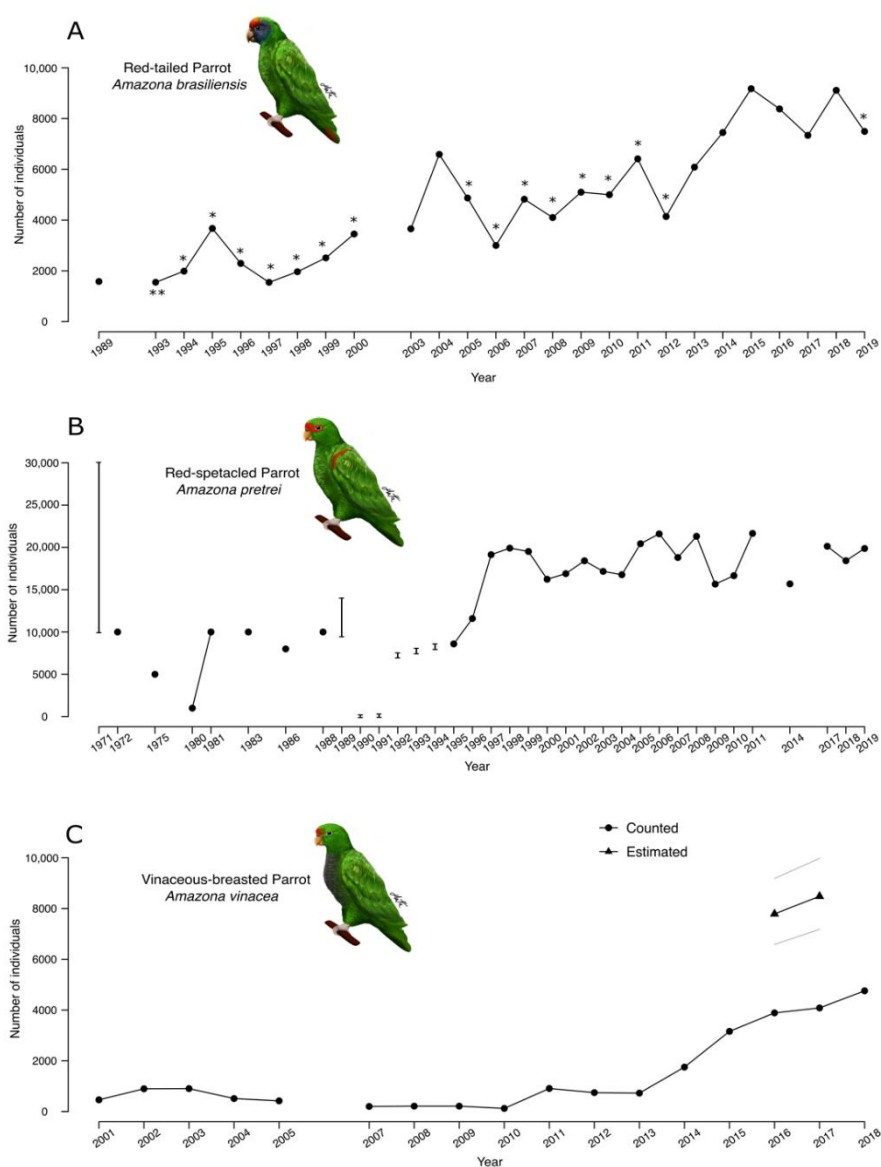

**Figure 3.** Number of Red-Tailed (**A**), Red-Spectacled (**B**), and Vinaceous-Breasted (**C**) Parrots counted by research teams throughout the last fifty years. Panel (**C**) also includes two estimates of the Vinaceous-Breasted Parrot population size, with gray lines showing bounds of the 95% credible interval of the a posteriori distribution of population size. These are the only statistical estimates of population size available in the literature for any of the study species. There is no plot for the Red-Browed Parrot because we could not find published records of count results for this species. Some of the Red-Spectacled Parrot counts were reported as intervals and appear as vertical lines in panel (**B**). Variations in the number of counted individuals may be due to variation in sampling effort or to real change in population size. Asterisks in panel (**A**) show differences in sampling effort: * corresponds to years that the counts were performed only in Paraná, and ** only in São Paulo. Sources for the numbers shown are [28,29,43–61] (**A**), [36–42,67] (**B**), and [21,27,42,62–64] (**C**).

## 4. Discussion

### 4.1. Geographic Range

The four *Amazona* species we studied showed marked differences in geographic range and, most likely, also in population size. The estimated areas of the geographic ranges varied over two orders of magnitude, from the approximately fifteen thousand square kilometers in *A. brasiliensis* to more than four hundred thousand square kilometers in *A. vinacea*. The mean estimated range was larger than the IUCN Extant area for all species, with 95% credible intervals including the IUCN Extant area for only one of them, *A. brasiliensis*. Both our estimated range area and the IUCN Extant areas approximate areas of occupancy as defined by Gaston and Fuller [68]. The marked disparity is likely a reflection of conservative caution in the definition of IUCN Extant areas and of extraordinary sampling coverage afforded by the use of citizen-science data in our estimates.

Geographic-range differences across species partially reflect environmental factors that limit their distribution. The range of *A. brasiliensis* appears to be limited by the highlands of the Serra do Mar [28], which also have high occupancy by *A. vinacea*. Indeed, *A. vinacea* is the only species to show evidence of a positive association between altitude and occupancy probability. Occupancy by *A. vinacea* is also positively associated with Atlantic and Araucaria Forest covers, even though the parrot's range extends beyond that of *Araucaria angustifolia* [17]. None of these associations—with altitude or with any type of forest cover—were evident from the analyses of the other three species—*A. brasiliensis*, *A. pretrei*, or *A. rhodocorytha*. Such lack of statistical association does not mean that they are biologically indifferent to forest cover. They are all cavity-nesters, and will not reproduce without access to tree holes, which are predominantly found in old-growth forests [28,29,66,69–71]. Instead, the focal areas of all three include extensive regions of forest (or of high or low altitude) that happen to not be occupied. This weakens the statistical association with occupancy covariates, not because they do not facilitate occupancy, but because unknown factors not included in our models may be further restricting the parrot distributions.

### 4.2. Population Size

Of the four species in this study, we only have a statistical estimate of global population size for *A. vinacea*. At around 8500 individuals [21], this estimate is nearly three times the number reported by the IUCN [72]. The local estimate of ~6000 *A. rhodocorytha* individuals for one 461 km$^2$ site in Espírito Santo, reported by Marsden et al. in 1998 [62], appears too high. This number, which implies a homogeneous density of 13 individuals per km$^2$ throughout the study site, is more than twice the number counted for the whole state of Espírito Santo by a different team ten years later [63]. There was either a dramatic population reduction in the state or these *A. rhodocorytha* numbers need reconsideration. There are no published estimates or counts of *A. rhodocorytha* for five of the states covered by the range map in Figure 1C. The species' global population size of 10,000 individuals reported by the IUCN [73] and the Brazilian Red List [74] may be reasonable, but neither source provides an explanation of how that number was obtained.

Any considerations about population sizes of *A. pretrei* and *A. brasiliensis* must be based solely on raw counts, as there are no published statistical estimates of population size for these species. Counts are difficult to compare because they do not quantify uncertainty about their values. They are also likely to underestimate real population size because they do not account for detection errors. In the absence of statistical estimates, however, counts offer a reasonable lower bound for population size. *A. pretrei* is the species with the largest counts, exceeding 20,000 individuals in 2006, 2008, and 2011, a number that is also greater than the 16,000 individuals cited by the species' IUCN Red List profile [75,76]. This species' well-known tendency to concentrate in only a few municipalities during the non-breeding season [39] reduces the probability that observers overlook large flocks and makes us relatively more confident of the accuracy of *A. pretrei*'s counts than of the others. Counts of *A. brasiliensis* reached more than 9000 individuals in 2018 [61,77], making it, possibly,

the species with the smallest geographic range but the second highest population size in this study. Future research could be aimed at the question of whether *A. brasiliensis* presents an exception to the well-supported positive relationship between area of occupancy and local abundance [78].

*4.3. Seasonal Change in Geographic Range*

Seasonal movements of aggregation and dispersion, influenced by the reproductive cycle and changes in food availability, are well-documented for *A. brasiliensis* [29], *A. pretrei* [39], and *A. vinacea* [21,27]. Dispersion occurs in the beginning of the breeding season (August–September), when breeding pairs abandon collective roosts to start spending the nights near the nest. By the end of the breeding season—December to March depending on the species—parrots aggregate again in collective roosts, which vary in size from dozens to thousands of individuals [21,27,29,39,79]. Aggregation and dispersal phases of *A. pretrei* occur in nearly non-overlapping parts of the species' range. By early Autumn, individuals concentrate in southeast Santa Catarina [39,80], and they spend the coolest months of the year in this region, feeding on abundant Paraná pine (*Araucaria angustifolia*) seeds [39] while other food resources are scarce [26]. Even though some individuals may overwinter in Rio Grande do Sul, the majority of the *A. pretrei* population spends this period in Santa Catarina, forming groups with thousands of individuals, in the municipalities of Painel, Urupema, Lages, and São Joaquim [39]. Between July and September, *A. pretrei* individuals disperse back to breed in Rio Grande do Sul, reaching at this point their largest geographic range and smallest group sizes [39]. Providing evidence of range dynamics at a larger temporal scale, *A. pretrei*'s center of aggregation has not always been in southeast Santa Catarina. Reports from the 1970s show large wintering aggregations of more than 10,000 individuals in the municipality of Muitos Capões, northern Rio Grande do Sul [3,37,38,67]. By the early 1990s, however, this number had decreased to only a few tens of individuals [39], and larger groups began appearing in Southeast Santa Catarina [39,81]. This shift of more than 100 km to the north follows decades of intense exploration and widespread destruction of Paraná pine forests in RS, which peaked between the 1920s and 1950s [82]. Most likely, scarcity of their most important winter fallback food forced *A. pretrei* into the colder but still relatively abundant Araucaria forests of the new wintering grounds in Santa Catarina [81].

*4.4. Long-Term Changes in Geographic Range*

Among the four species in our study, *A. vinacea* shows the strongest evidence of range contraction, with local extinctions in parts of Argentina and Paraguay since the 1970s [63,64]. With a historic range that covered southern Paraguay west of Misiones and all the way into central Paraguay to the northwest [63], the occurrence of *A. vinacea* outside Brazil is now restricted to three localities in Argentina [21,63,64] and two in Paraguay [21,63]. Both *A. vinacea* and *A. pretrei* are classified as critically endangered in Argentina [83], which may have had a historical population of the latter [84,85] *A. pretrei* is rarely observed in Paraguay [23,86], where it is also classified as threatened [87]. Belton [38] mentions the possible past occurrence of *A. pretrei* as far north as São Paulo state, in Brazil, but the validity of XIX century records that could backup such possibility is disputed [3,75]. Reviewing information about *A. brasiliensis*, Scherer-Neto [28] cites reports of XIX century sightings in northern Rio Grande do Sul and northeast Santa Catarina (see also [77]), but the validity of these reports, too, is questionable [3]. Even with reliable identification, though, past observation of any species far outside the present range is no firm evidence of range contraction. Individuals may wander away from their species' ranges, sometimes across oceans [88], with sightings in unexpected locations inevitably getting more attention than within a known range, even if they bear no consequence to population dynamics. Parrots introduce the additional complication of having been kept as pets for a long time, so that past sightings in odd places could also be of individuals escaped from captivity. To conclude, *A. rhodocorytha* has the least historical information of the four species, with perhaps

one observation deserving special attention: one recent record in the state of Alagoas [89] dispels a previous suggestion of local extinction [3] and confirms the existence of a disjunct population in the extreme north of the distribution.

### 4.5. Long-Term Change in Population Size

The time series of counts that we report for *A. vinacea* and *A. brasiliensis* show increasing numbers very likely due to an increase in sampling effort. The time series for *A. pretrei* shows relatively small variation for the last two and a half decades. After an apparent decline during the 1970s [37,38], *A. pretrei* counts increased to around 20,000 individuals in 1997. Such increase coincides with the period when *A. pretrei* was shifting its wintering aggregation to Southeast Santa Catarina, where counts have been carried out by the same research team since 1995. Counts of *A. brasiliensis* and *A. vinacea*, on the other hand, have been carried out by different research groups in different locations, with variable degrees of coordination. The highest counts of *A. brasiliensis*, for example, were obtained in 2015 (9176 individuals), and in 2018 (9112 individuals), when research teams visited all known roosts in São Paulo and Paraná. In 2019, however, when only Paraná roosts were visited, approximately 2000 fewer individuals were counted. Similar, effort-driven variation is evident in the *A. vinacea* time series, which had fewer than one thousand individuals counted annually from 2001 to 2013. *A. vinacea* counts have increased since 2014, with the implementation of annual coordinated counts performed at a number of sites, that increased gradually from 20, in 2014, to 67, in 2017. The only period for which we can draw statistical inference about temporal change in the *A. vinacea* population is the transition from 2016 to 2017 [21]. The estimates shown in Figure 3C account for detection error and for variation in effort between the two years. The credible intervals of the abundance estimates, broadly overlapping between the two years, provide no evidence of a substantial change. Future analysis of population trends will require more coordination and replication of counts. This will facilitate statistical analysis of count results and investigation of real trends in population size.

### 4.6. Concluding Remarks

The future of the four parrot species analyzed in this study is threatened by two key environmental hazards: habitat loss and human exploitation [2]. *A. brasiliensis*, *A. pretrei*, *A. rhodocorytha*, and *A. vinacea* are all impacted by the destruction of the Atlantic Forest, especially because they nest in tree cavities that are much more common in old growth than in secondary forests [90]. Since the arrival of Europeans in South America, almost 90% of the original Atlantic Forest cover was lost [12]. The remaining forest is highly fragmented, with only 20% of its area contained in patches larger than 100 km$^2$, and 83% of the patches being smaller than 50 hectares [12]. When not replaced by pasture or farmland, cleared forest gives way to exotic tree monocultures, such as *Pinus* and *Eucalyptus* [91]. In coastal areas intensely used by the tourism industry, cleared forests may also give way to urban expansion, which disproportionally affects *A. brasiliensis* [29]. The other primary threat to all four species, human exploitation, comes in the form of nest poaching [8,69,74,92–96]. According to one study [8], nest poaching is the principal cause of nest failure for *A. vinacea* —with more than 80% of 25 monitored nests poached—and *A. brasiliensis*—with 50% of 78 monitored nests poached. Four initiatives have been promoting conservation, as well as research and monitoring of the four species throughout the last three decades: Projeto Charão (for *A. pretrei*, since 1991), Projeto para Conservação do Papagaio-de-cara-roxa (for *A. brasiliensis*, since 1997), Projeto Chauá (for *A. rhodocorytha*, since 2014), and the Programa Nacional para a Conservação do Papagaio-de-peito-roxo (for *A. vinacea*, since 2015). To improve knowledge about population dynamics and manage a response to environmental threats, it is essential that these and similar initiatives expand their reach. Continued tapping of citizen-science data will help to update knowledge about species' ranges. The estimation of abundance and validation of range maps, however, require observers on the ground. Much can be achieved just by sending observers to municipalities with higher un-

certainty about species' occurrence, but one can go much further by practicing integration of citizen-science and professional research work on a routine basis. The combination of range mapping based on citizen-science and coordinated observation by research teams throughout the species' ranges offers a powerful tool for accurately monitoring the species' status and for assessing the consequences of management decisions.

**Supplementary Materials:** The following are available online at https://www.mdpi.com/article/10.3390/d13090416/s1, Supplementary S1: The R and BUGS code for the models used in estimating the parrot's geographic range is available. Table S1: Compilation of the available counts for Amazona brasiliensis with the respective reference. Table S2: Compilation of the available counts for Amazona pretrei with the respective reference. Table S3: Compilation of the available counts and abundance estimates for Amazona vinacea with the respective reference.

**Author Contributions:** Conceptualization, V.Z. and G.F.; methodology, D.A.W.M.; software, D.A.W.M. and V.Z.; formal analysis, V.Z. and D.A.W.M.; investigation, V.Z. and G.F.; resources, V.Z., D.A.W.M. and G.F.; data curation, V.Z.; writing—original draft preparation, V.Z., D.A.W.M. and G.F.; supervision, G.F.; project administration, V.Z. and G.F.; funding acquisition, V.Z. and G.F. All authors have read and agreed to the published version of the manuscript.

**Funding:** V.Z. received fellowships from the Brazilian government's CAPES and PrInt/CAPES programs, as well as research funding from Funbio, Instituto Humanize, and The Rufford Foundation (19835-1 and 23709-2).

**Institutional Review Board Statement:** Not applicable.

**Informed Consent Statement:** Not applicable.

**Data Availability Statement:** The R codes and datasets used for the analyses are openly available at GitHub at: https://github.com/vivizulian/DataIntegrationModels (accessed on 3 April 2021).

**Acknowledgments:** We are grateful to Reinaldo Guedes who volunteered his free time over the last twelve years to developing and maintaining WikiAves, the most successful citizen-science initiative in Brazil. This paper would not be possible without the help of thousands of bird observers who uploaded photos, audio recordings, and birding lists to eBird, WikiAves, and Xeno-canto. Glayson Bencke kindly shared his time and knowledge about parrot biology with us in several insightful conversations that contributed to shaping the best part of this paper. We also thank the three reviewers that helped improve the manuscript.

**Conflicts of Interest:** The authors declare no conflict of interest.

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
