# Peer review of "Endemic and Threatened Amazona Parrots of the Atlantic Forest: An Overview of Their Geographic Range and Population Size"

_diversity, doi:10.3390/d13090416_

Round 1

Reviewer 1 Report

This is valuable contribution to the conservation assessment, and conservation, of these parrots.

I feel you’ve misunderstood the relationship between IUCN metrics and geographic range size. EOO does not measure range size in any sense near what you’ve measured, but rather the geographic spread of risk. I urge you to read Gaston & Fuller (2009) and remove the emphasis in the 2nd paragraph of Results and again in the Discussion – where I note you’ve at least partially grasped the issue and acknowledged the pointlessness of the comparison. If you wish to make a comparison with the IUCN assessments, then assess EOO (convex polygon) directly from your records.

I found your M&M section difficult to follow because it doesn’t always follow a logical sequence. For example, in the early paragraphs about your general process and sources of field data there are statements (l90 et al. and again l109-112) about data analysis. I think the section would be easier to read with shorter paragraphs isolating distinct topics, and sub-headings along the lines of “Parrot records” and “Data analysis”.

Your Discussion is rather too long, I feel, and not so well focussed on your Results. It might be improved with identification of several discrete topics and use of sub-headings within which to present them. I also feel that your discussion of threats could be improved with respect to habitat loss – you appear (l474) to interpret loss of habitat almost solely in terms of loss of nest hollows. What about loss of feeding areas that may be seasonally important? Is there any evidence to support the notion that hollows are more critical than feeding grounds?

Your use of English is generally good, but significant issues remain that require attention. I’ve provided some suggestions for the Abstract and Introduction only; the entire manuscript needs to be reviewed to tighten presentation in this way.

Minor points:

l37-8, “highly dependent on forest habitats”. This is a considerable over-generalisation. Australian parrots, at least, occupy a much wider range of habitats including habitat specialists in heath, grasslands, woodlands, tropical savannas, semi-desert scrubs and deserts. There are also many species that have adopted urban and rural areas as habitat. I believe this to be at least somewhat true for African parrots as well.

l48: why “inevitably”?

l199: “estimate. respective” ???

l203-13: this paragraph essentially repeats Table 1 and contributes little. It could be much reduced.

l214-5: EOO surely stands for Extent of Occurrence and, so far as I can see, “Estimated” doesn’t explicitly form part of the IUCN’s term.

l218: surely you mean EOO not EEO

l272: “posterior”? I suggest deleting this word.

l299: for consistency and clarity, replace “Red-browed Parrot” with “A. rhodocorytha

l312: why should we anticipate that “phylogenetic proximity and similar appearance” would yield similar range size, and thus why is finding of dissimilarity worth reporting in this way?

Examples where text could be improved – Abstract and Introduction only

Abstract, 1st sentence: this is awkward with too many clauses separated by commas

l30-34: sentence too long and awkward with too many clauses separated by commas

l51-2: replace “one of the global hotspots of biodiversity” with “a global biodiversity hotspot”

l52: a biome doesn’t lose: re-word

l54: replace “one projection [13], until the year 2070” with “one projection to 2070 [13]”

l55: a region doesn’t “suffer”; re-word

l56: delete “decadal bird habitat” [repetitive]

l57: replace “Atlantic Forest land use” with “land use in the Atlantic Forest”

l66-7: replace “the two state variables of geographic range and population size” with “these two variables” [repetitive]

l78: replace “when” with “where”

l81: delete “accurate” [meaningless term]

Reference

Gaston KJ, Fuller RA. 2009. The sizes of species' geographic ranges. Journal of Applied Ecology 46: 1-9.

Author Response

Response to Reviewer’s comments on Manuscript ID: diversity-1267950

Comments are marked as ‘REV1’ as they were written by Reviewer #1. Our responses to comments are marked ‘R’ and numbered.

Reviewer 1:

Comments and Suggestions for Authors

REV1: This is valuable contribution to the conservation assessment, and conservation, of these parrots. I feel you’ve misunderstood the relationship between IUCN metrics and geographic range size. EOO does not measure range size in any sense near what you’ve measured, but rather the geographic spread of risk. I urge you to read Gaston & Fuller (2009) and remove the emphasis in the 2nd paragraph of Results and again in the Discussion – where I note you’ve at least partially grasped the issue and acknowledged the pointlessness of the comparison. If you wish to make a comparison with the IUCN assessments, then assess EOO (convex polygon) directly from your records.

R1: Thank you for pointing out this problem. We read Gaston & Fuller (2009) and now understand the importance of differentiating between EOO and AOO. Instead of trying to extract EOO values from our maps, however, we decided to extract something akin to AOO values from the area of the IUCN Extant polygons available for each species at the IUCN Red List website. We will call these ‘IUCN extant areas’. We would like to avoid making an arbitrary decision about the occupancy probability at which we draw the limits of a convex polygon over our maps. At the same time, we believe that it is useful to compare the area given by our new maps with the area represented in the best maps available to us at the beginning of our study.

We removed the EOO values and comments about EOO from the results and from the discussion.

REV1: I found your M&M section difficult to follow because it doesn’t always follow a logical sequence. For example, in the early paragraphs about your general process and sources of field data there are statements (l90 et al. and again l109-112) about data analysis. I think the section would be easier to read with shorter paragraphs isolating distinct topics, and sub-headings along the lines of “Parrot records” and “Data analysis”.

R2: We inserted two sub-headings (‘Study area and data collection’ and ‘Data analysis’) to make the M&M section easier to follow. To avoid mixing data sourcing and analysis information, we transferred the sentences pointed out by the reviewer to what is now ‘Data analysis’ part of the M&M section.

REV1: Your Discussion is rather too long, I feel, and not so well focused on your Results. It might be improved with identification of several discrete topics and use of sub-headings within which to present them. I also feel that your discussion of threats could be improved with respect to habitat loss – you appear (l474) to interpret loss of habitat almost solely in terms of loss of nest hollows. What about loss of feeding areas that may be seasonally important? Is there any evidence to support the notion that hollows are more critical than feeding grounds?

R3: We unfortunately do not have information about the relative importance of nest hollows and critical feeding grounds for the population growth, but we did breakdown the discussion section with sub-headings.

REV1: Your use of English is generally good, but significant issues remain that require attention. I’ve provided some suggestions for the Abstract and Introduction only; the entire manuscript needs to be reviewed to tighten presentation in this way.

R4: Thank you. We hope that the new version is sufficiently improved.

Minor points:

REV1: l37-8, “highly dependent on forest habitats”. This is a considerable over-generalisation. Australian parrots, at least, occupy a much wider range of habitats including habitat specialists in heath, grasslands, woodlands, tropical savannas, semi-desert scrubs and deserts. There are also many species that have adopted urban and rural areas as habitat. I believe this to be at least somewhat true for African parrots as well.

R5: We agree that some parrot species are not dependent from the forest habitats. However, according to the IUCN Red List, 94% (373) of the Psittacidae species use forest habitats, which implies a considerable level of dependence. We removed the word ‘highly’.

REV1: l48: why “inevitably”?

R6: Word removed.

REV1: l199: “estimate. respective” ???

R7: This sentence was misplaced and we excluded the sentence from the methods. Thank you.

REV1: l203-13: this paragraph essentially repeats Table 1 and contributes little. It could be much reduced.

R8: We agree and shortened the paragraph by 32 words.

REV1: l214-5: EOO surely stands for Extent of Occurrence and, so far as I can see, “Estimated” doesn’t explicitly form part of the IUCN’s term.

R9: Yes, we made a mistake here. As explained in R1, above, we now removed all the comments about extent of occurrence.

REV1: l218: surely you mean EOO not EEO

R10: Yes, correct. We now removed every mention of EOO.

REV1: l272: “posterior”? I suggest deleting this word.

R11: We agree and deleted the word.

REV1: l299: for consistency and clarity, replace “Red-browed Parrot” with “A. rhodocorytha

R12: Of course, thank you. Common name replaced by Latin name on line 449.

REV1: l312: why should we anticipate that “phylogenetic proximity and similar appearance” would yield similar range size, and thus why is finding of dissimilarity worth reporting in this way? 

R13: This is a good point. We do not have solid information to support our statement and therefore decided to exclude it.

Examples where text could be improved – Abstract and Introduction only:

REV1: Abstract, 1st sentence: this is awkward with too many clauses separated by commas

R14: Sentence split and rephrased.

REV1: l30-34: sentence too long and awkward with too many clauses separated by commas

R15: Sentence reformulated on lines 32-33.

REV1: l51-2: replace “one of the global hotspots of biodiversity” with “a global biodiversity hotspot”

R16: OK, replacement done on line 66.

REV1: l52: a biome doesn’t lose: re-word

R17: Sentence re-worded, lines 67.

REV1: l54: replace “one projection [13], until the year 2070” with “one projection to 2070 [13]”

R18: OK, changed on line 69.

REV1: l55: a region doesn’t “suffer”; re-word

R19: OK, sentence reformulated on lines 69-70.

REV1: l56: delete “decadal bird habitat” [repetitive]

R20: Deleted on line 70.

REV1: l57: replace “Atlantic Forest land use” with “land use in the Atlantic Forest”

R21: Done, on line 71.

REV1: l66-7: replace “the two state variables of geographic range and population size” with “these two variables” [repetitive]

R22: OK, replaced on line 79.

REV1: l78: replace “when” with “where”

R23: Replaced on line 90.

REV1: l81: delete “accurate” [meaningless term]

R24: Deleted from line 93.

REV1: Reference

Gaston KJ, Fuller RA. 2009. The sizes of species' geographic ranges. Journal of Applied Ecology 46: 1-9.

R25: Thank you!

Reviewer 2 Report

General comments:

This study examined patterns of parrot occurrence abundance in the Brazilian Atlantic Forest, using a combination of data sets from empirical studies and citizen science, and for occupancy they analyzed these using a novel application of a data integration approach. Improved estimates of distribution for species such as these are timely and needed given their conservation plight. Therefore, the study is well justified and well written for the most part. However, more clarity is needed in describing their occupancy modeling approach, and I struggled with the relevance of displaying trends in counts over time. From what I could tell there was very little detail in the methods on where and how these counts were obtained (except for A. vinacea). As the authors readily acknowledged, they often couldn’t account for differences in effort, and therefore I worry that even with that caveat, comparing trends over time or by species could be misleading. I offer specific comments below.

Specific comments:

Lines 98-99: Is Zulian et al. (2021) the biorxiv preprint?

Lines 103-105: What’s the significance of this 11-year time period?

Lines 121-123: Can you clarify what 154 represents? My guess is the standard deviation, but I doubt there are negative numbers of checklists. Why not just report a range?

Lines 123-125: I don’t understand, what do the WikiAves and Xeno-canto data represent? Do they simply say whether a species was in a municipality or not? You state that they gather records for a municipality, what are those records? If a species is detected in one record but not another, is that not like eBird? Please clarify.

Lines 157-165: Were any of these covariates highly correlated, if you include them in the same model?

Lines 176-177: Why did you use the 0.5 degree size for your cells?

Lines 183-185: This method should be explained and justified more here. You included a more detailed explanation in the Zulian et al. preprint, but since that has not undergone peer review (as far as I know), it would be good to include your explanation here too.

Line 186: What does DS1 represent? One of the datasets?

Lines 199-201: I think you are missing the first part of this sentence.

Lines 225-226: These “slight” effects on occupancy do not appear to be supported in Table 2, since credible intervals overlapped 0.

Lines 252-254: You interpret these occupancy values relative to 0.1, yet in Fig. 1 you interpret z~0.5 as indicating high uncertainty, so perhaps it would be better to interpret the number of municipalities with z closer to 1?

Lines 268-273: Summaries such as these make me wonder if they are useful, if you cannot account for differences in effort and monitoring location among these studies.

Lines 277-280: Again, as you concede, trends in counts are confounded by differences in effort, so why present these numbers as a trend?

Lines 316-318: Falling out how, and which 95% credible interval of the estimate? Please clarify this sentence.

Lines 371-376: It is not clear to me how you are obtaining a density estimate from these counts.

Lines 446-470: Shouldn’t this paragraph be combined with the paragraphs above (lines 347-387)? The discussion seems to jump between occupancy and abundance but it is not clear why.

Data availability Statement: It would be nice to at least be able to review the code behind these models.

Table 2. Please check your estimate for effect of Altitude on A. vinacea.

Author Response

Response to Reviewer’s comments on Manuscript ID: diversity-1267950

Comments are marked as ‘REV2’ as they were written by Reviewer #2. Our responses to comments are marked ‘R’ and numbered.

Reviewer 2

Comments and Suggestions for Authors

General comments:

REV2: This study examined patterns of parrot occurrence abundance in the Brazilian Atlantic Forest, using a combination of data sets from empirical studies and citizen science, and for occupancy they analyzed these using a novel application of a data integration approach. Improved estimates of distribution for species such as these are timely and needed given their conservation plight. Therefore, the study is well justified and well written for the most part. However, more clarity is needed in describing their occupancy modeling approach, and I struggled with the relevance of displaying trends in counts over time. From what I could tell there was very little detail in the methods on where and how these counts were obtained (except for A. vinacea). As the authors readily acknowledged, they often couldn’t account for differences in effort, and therefore I worry that even with that caveat, comparing trends over time or by species could be misleading. I offer specific comments below.

R1: Thank you for the comments. We added more information about the modelling approach on lines 277-294. We also struggled with the decision of showing time series of counts, but in the end, we concluded that showing these numbers is better than omitting the information altogether. Counts offer reasonable lower bound for population size and the variation in the length of the time series gives an illustration of what species have received more or less attention from the research community. We added some information about how counts were obtained on lines 100-103 and a table in the Supporting Information with all the counts and the reference from where each count was extracted (Table S1).

Specific comments:

REV2: Lines 98-99: Is Zulian et al. (2021) the biorxiv preprint?

R2: Yes, it is. This manuscript is also on its second round of review at Diversity and Distributions.

REV2: Lines 103-105: What’s the significance of this 11-year time period?

R3: We initially intended to analyze 10-years of data up to 2018. We felt that much more than 10 years would be too long for analyzing with a static model. However, when we realized that WikiAves was launched in 2008, the same year that eBird started to gain popularity in Brazil, we decided to analyze the whole period of available data, from 2008 to 2018, which returns an 11-year period.

REV2: Lines 121-123: Can you clarify what 154 represents? My guess is the standard deviation, but I doubt there are negative numbers of checklists. Why not just report a range?

R4: The reviewer is correct. That number is the standard deviation of the number of lists per municipality. Note, however, that the distribution of lists per municipality is not symmetric so the high standard deviation does not imply negative numbers of checklists. In any case, to avoid confusion we replaced this number by the minimum and maximum number of lists per municipality on lines 161 and 162.  The new sentence reads: “The number of lists per municipality varied from 1 to 3,245, with a mean of 33 lists, collected at different times of the year by different observers.”

REV2: Lines 123-125: I don’t understand, what do the WikiAves and Xeno-canto data represent? Do they simply say whether a species was in a municipality or not? You state that they gather records for a municipality, what are those records? If a species is detected in one record but not another, is that not like eBird? Please clarify.

R5: Unlike eBird that records lists of species, Wikiaves and Xeno-canto record observer input in the form of individual photos or audio recordings of an identified species. Considering that these individual records do not have a direct effort measure, we used the total number of photos and the total number of audio recordings of any species from each municipality as effort covariates for these two datasets. This mean that we have a vector of detection/non-detection data for each parrot species, with length equal to the number of municipalities and values of ‘1’ or ‘0’, respectively, for those municipalities that did or did not have at least one photo or audio recording of the target species.

REV2: Lines 157-165: Were any of these covariates highly correlated, if you include them in the same model?

R6: No, the covariates are not correlated. We checked for correlation before including the variables in the models.

REV2: Lines 176-177: Why did you use the 0.5 degree size for your cells?

R7: We fitted the Full model with three different hexagon widths in the CAR component: 0.25°, 0.5° and 1° latitude. Because total deviance was smaller with 0.5° we kept that hexagon size for all the analyses reported in the paper.

REV2: Lines 183-185: This method should be explained and justified more here. You included a more detailed explanation in the Zulian et al. preprint, but since that has not undergone peer review (as far as I know), it would be good to include your explanation here too.

R8: We rewrote this section adding more detailed information about the effort equations. See lines 277-294.

REV2: Line 186: What does DS1 represent? One of the datasets?

R9:  represents the Dataset 1. It was the way we found to clarify that the effort equation (Eq. 4) is used with all datasets, one at a time. To clarify, we changed  for  in Equation 4 and added the information that  varies between 1 and 4 on line 280.

REV2: Lines 199-201: I think you are missing the first part of this sentence.

R10: Yes, thank you. The sentence was related with the results, so we excluded it from the methods section.

REV2: Lines 225-226: These “slight” effects on occupancy do not appear to be supported in Table 2, since credible intervals overlapped 0.

R11: The reviewer is correct. These effects are not sufficiently supported by the data. We excluded the sentence on line 329.

REV2: Lines 252-254: You interpret these occupancy values relative to 0.1, yet in Fig. 1 you interpret z~0.5 as indicating high uncertainty, so perhaps it would be better to interpret the number of municipalities with z closer to 1?

R12: We agree that 0.1 is too low. We rewrote the sentences on lines 394-398 accounting for the number of municipalities with z > 0.9 (which are represented by the darker tone of red in Figure 1).

REV2: Lines 268-273: Summaries such as these make me wonder if they are useful, if you cannot account for differences in effort and monitoring location among these studies.

R13: We agree that this is a difficult choice, but we still believe that presenting raw counts will be better than ignoring the information altogether. See R1, above. In the manuscript, we tried to make it clear that we urgently need replicated counts in order to obtain estimates of population size.

REV2: Lines 277-280: Again, as you concede, trends in counts are confounded by differences in effort, so why present these numbers as a trend?

R14: We were very careful to avoid any mention of a ‘trend in counts’ in the text. Lines 637-639 state “Future analysis of population trends will require more coordination and replication of counts. This will facilitate statistical analysis of count results and investigation of real trends in population size.”

REV2: Lines 316-318: Falling out how, and which 95% credible interval of the estimate? Please clarify this sentence.

R15: We reworded to: “The mean estimated range was larger than the IUCN ‘Extant’ area for all species; the 95% credible interval of the estimated range included the IUCN ‘Extant’ area for only one species, A. brasiliensis.”

REV2: Lines 371-376: It is not clear to me how you are obtaining a density estimate from these counts.

R16: It is a density of animals counted per unit area of the range. We just divide the maximum count of each species by the estimated area of the range. We inserted a short explanation on lines 534-535.

REV2: Lines 446-470: Shouldn’t this paragraph be combined with the paragraphs above (lines 347-387)? The discussion seems to jump between occupancy and abundance but it is not clear why.

R17: We apologize for the confusion here. Our intention was to address the following the topics (in the following order): static geographic range, static population size, temporal change in geographic range, and temporal change in population size. We added four subheadings in the discussion to clarify.

REV2: Data availability Statement: It would be nice to at least be able to review the code behind these models.

R18: Yes, of course! We added a Supplemental material with the code in the new version of the manuscript (Appendix S1).

REV2: Table 2. Please check your estimate for effect of Altitude on A. vinacea.

R19: Thank you! We corrected the estimate.

Reviewer 3 Report

The study authored by Zulian et al. performed a geographical and population size survey on four Amazona parrot species, whereas the parrots are the endemic, threatened and red lists under the IUCN. Further, the authors have brief introduction about the IUCN, parrot species, and the present study objectives on the introduction part. The dataset collection and the statistical analysis were conducted well.

However, Few corrections and clarification are necessary for the improvement of the article.

Minor corrections

1.The authors have investigated the population size and geographical survey in the four parrot species. The present title looks like the review article. Hence, it would be appropriate to include a word "survey" in the title and change the title according to the study.

2.Materials and methods, line 165 : provide the database link (https://www.ibge.gov.br/) in the article.

3.Line 166: Provide this reference for reference 30 (DIVA-GIS) and give the website link (https://www.diva-gis.org/) in the main text. 
Hijmans, R. J., Guarino, L., Cruz, M., & Rojas, E. (2001). Computer tools for spatial analysis of plant genetic resources data: 1. DIVA-GIS. Plant genetic resources newsletter, 15-19.

4.Line 194: give an expansion for MCMC "(Markov Chain Monte Carlo) algorithm". 

5.Result part, Line  218: The authors have indicated IUCN "EEO's", what does mean? is it "Estimated extent of occurrence EOO"?  EEO or EOO? Verify and provide correct term in whole manuscript. Beacuse, mentioning different in the discussion part (line 318).

6.Result part, line 295: Marsden et al., [8] not in the reference section. The author needs to verify all the reference cited in the main text.

7.Verify the reference cited for line 300, not relevant to the sentence. Because, we couldn't identify the research work on Amazona rhodocorytha in the cited article (Tella and Hiraldo, 2014).

8.Table 2. Remove quotation ('), change to "each species model".

Major comments

1. Introduction part: missing impact on conservation and how this present geographical and population surveys useful for the future research. The author needs to include a sentence in the introduction part.

2. Cite the article in relevant to the manuscript and write a sentence in the discussion part (https://onlinelibrary.wiley.com/doi/abs/10.1111/jofo.12256). I suggest the authors to include the few points in the intro or discussion part to strengthen this manuscript further.

3. Why the estimated abundance was not accessed in other three species except A. vinacea?

4. It will be interesting to see the diversity phylogenetic tree of the amazona species (without redundant of 36 species or 25 red list species) and how these 4 described species dispersed in evolutionary distances? Add the dendogram view of phylogenetic tree in the supplementary information.

5.The present study was carried out from the consolidated dataset of previously observed dataset of other database (ebird, wikiaves, xenocanto) and the authors performed only geographical map and statistical analysis. 

6.Several articles references were cited in another language (notably 25-27;35, 40, 58-60;64, 66, 74, 76-77, 80, 84, 86, 89, 92-94 ) (not in english, may be in Portugais), so difficult to check the references according to the citation in the manuscript.

Author Response

Response to Reviewer’s comments on Manuscript ID: diversity-1267950

Comments are marked as ‘REV3’ as they were written by Reviewer #3. Our responses to comments are marked ‘R’and numbered.

Reviewer 3

Comments and Suggestions for Authors

REV3: The study authored by Zulian et al. performed a geographical and population size survey on four Amazona parrot species, whereas the parrots are the endemic, threatened and red lists under the IUCN. Further, the authors have brief introduction about the IUCN, parrot species, and the present study objectives on the introduction part. The dataset collection and the statistical analysis were conducted well. However, few corrections and clarification are necessary for the improvement of the article.

R1: We thank to the reviewer for the encouraging comments.

Minor corrections

REV3: 1. The authors have investigated the population size and geographical survey in the four parrot species. The present title looks like the review article. Hence, it would be appropriate to include a word "survey" in the title and change the title according to the study.

R2: We beg to differ here. In our understanding the word ‘survey’ conveys an idea of a more thorough scrutiny than ‘overview’. We believe that we are being thorough when it comes to the evaluation of geographic ranges; but when it comes to population size, we really don’t have enough information to claim that we did something as thorough as a survey. We would really like to keep the original title, if that is acceptable.

REV3: 2. Materials and methods, line 165: provide the database link (https://www.ibge.gov.br/) in the article.

R3: We added the link on line 258.

REV3: 3. Line 166: Provide this reference for reference 30 (DIVA-GIS) and give the website link (https://www.diva-gis.org/) in the main text. 
Hijmans, R. J., Guarino, L., Cruz, M., & Rojas, E. (2001). Computer tools for spatial analysis of plant genetic resources data: 1. DIVA-GIS. Plant genetic resources newsletter, 15-19.

R4: We added the link on lines 259-260 and corrected the reference.

REV3: 4. Line 194: give an expansion for MCMC "(Markov Chain Monte Carlo) algorithm". 

R5: We added the expansion on line 301.

REV3: 5. Result part, Line 218: The authors have indicated IUCN "EEO's", what does mean? is it "Estimated extent of occurrence EOO"?  EEO or EOO? Verify and provide correct term in whole manuscript. Because, mentioning different in the discussion part (line 318).

R6: We recognize that we made a confusion about EEO and EOO, but we now removed EOO from the paper. While addressing a comment by Reviewer 1, we replaced EOO information with the area extracted from IUCN ‘Extant’ polygons of each species. We now compare our estimates of geographic range size with the area of the IUCN extant polygons.

REV3: 6. Result part, line 295: Marsden et al., [8] not in the reference section. The author needs to verify all the reference cited in the main text.

R7: Thank you. We verified all the references now.

REV3: 7. Verify the reference cited for line 300, not relevant to the sentence. Because, we couldn't identify the research work on Amazona rhodocorytha in the cited article (Tella and Hiraldo, 2014).

R8: Thank you. We apologize for this mistake. The reference is now corrected.

REV3: 8. Table 2. Remove quotation (‘), change to “each species model”.

R9: Quotation removed.

Major comments

REV3: 1. Introduction part: missing impact on conservation and how this present geographical and population surveys useful for the future research. The author needs to include a sentence in the introduction part.

R10: We express relevance for conservation in the sentence “improved knowledge about abundance and distribution will help directing future monitoring and conservation efforts, as well as strengthening the basis for threat assessments.”.

REV3: 2. Cite the article in relevant to the manuscript and write a sentence in the discussion part (https://onlinelibrary.wiley.com/doi/abs/10.1111/jofo.12256). I suggest the authors to include the few points in the intro or discussion part to strengthen this manuscript further.

R11: We added this reference in the line 75 of the manuscript.

REV3: 3. Why the estimated abundance was not accessed in other three species except A. vinacea?

R12: Statistical estimation of abundance requires some form of sampling replication or distance sampling to allow for some correction of detection errors. The available count datasets for Amazona pretrei, A. brasiliensis and A. rhodocorytha do not include replication nor distance sampling. This is the reason we refer to the numbers for those species as ‘counts’ and not as ‘abundance’ through the manuscript.

REV3: 4. It will be interesting to see the diversity phylogenetic tree of the amazona species (without redundant of 36 species or 25 red list species) and how these 4 described species dispersed in evolutionary distances? Add the dendogram view of phylogenetic tree in the supplementary information.

R13: We agree that the understanding of the evolutionary history may help comprehends the actual distribution and population size of each species and we found this an interesting topic that should be explored in future research. We brought up phylogenetic thinking to the discussion in the first version, but following a commentary by Reviewer 1, we concluded that we did not have enough information or space to make informative statements about phylogeny here. We believe it is best to abstain from the topic in the scope of this paper. For this reason, we excluded the phrase about phylogenetic proximity of our target species in the first sentence of the discussion (line 465).

REV3: 5. The present study was carried out from the consolidated dataset of previously observed dataset of other database (ebird, wikiaves, xenocanto) and the authors performed only geographical map and statistical analysis. 

R14: We are not sure that we understood the issue that is being raised here. Our manuscript includes: 1) a compilation of all the available count data and abundance estimates, and 2) a statistical model of the geographic range of four species of parrots with high conservation importance. The count data compiled for three of the species does not include the minimum information needed for abundance estimation or trend analysis (see also R12, above).

REV3: 6.Several articles references were cited in another language (notably 25-27;35, 40, 58-60;64, 66, 74, 76-77, 80, 84, 86, 89, 92-94 ) (not in english, may be in Portugais), so difficult to check the references according to the citation in the manuscript.

R15: We understand that it is difficult to check when the references are in Portuguese, but these sources are essential for our work and we have no alternative but to cite them. There is no equivalent information published in English. Indeed, one of our motivations in writing this paper was to compile these papers, evaluate their information and summarize it in an English-language paper that could be accessible to a wide audience.

Round 2

Reviewer 1 Report

Your revision is a considerable improvement, and I’ve mostly only minor points of expression to suggest. However, I feel that your Discussion still leaves a deal to be desired.

Sub-headings in the Discussion certainly help, though these need some re-organisation and amendment. Your use of “Temporal” in sub-headings 4.3 and 4.4 is non-specific and confounds two very different issues. Also, the material in section 4.3 conflates seasonal change with long-term change. I recommend:
     4.3. Seasonal change
     4.4. Long-term trends
and moving relevant material from 4.3 to 4.4.
You have not shortened your Discussion – actually lengthened it a little – and it is often not well focussed on your findings. In a number of paragraphs there’s a confusing multiplicity of themes resulting in them being too long. For each paragraph, I urge you to identify its theme in a crisp first sentence (be quite specific to something arising directly from your results), then elaborate that theme and delete ALL extraneous material (if important, move the extraneous material to a new paragraph). Avoid sweeping generalities in first sentences – some might be moved to the end of paragraph to form a conclusion to it, but beware overstating your findings.
Throughout: I urge you to avoid incidentals and attempting to review every reference, instead sticking to key generalities relating directly to your results. If you feel all that detail is important to put on record, move it to Supplementary materials.
Here’s some specific issues:
- 4.1. second paragraph. So what about sympatry or lack thereof? (And it isn’t in your Results.) Delete first section of this paragraph and identify the paragraph theme (range relationships to forest cover) in the first sentence.
- 4.2. first paragraph (missing lines?): this paragraph has two distinct themes which are confusingly interwoven: actual population counts and estimates, and the problem with variation in survey effort. I recommend two paragraphs, the first dealing with counts and estimates, i.e. sentences on lines 499-503 and 509-528, the second cautioning with issues arising from variation in survey effort.
- 4.2. second paragraph (see also below). This makes much of little. Regardless of the generality outlined by Holt et al., I can think of so many exceptions that the point is trivial unless specifically framed in terms of contrasting ecologies – which is well beyond the scope of your paper. At the very least, this paragraph would benefit from considerable shortening. Further, the first sentence is confusing. Do you mean “Density estimates will vary with estimates of both population and geographic range.”? But actually I feel that the first two sentences could and should simply be deleted. Certainly, they don’t belong at the beginning of the paragraph because the paragraph topic is the “interesting pattern”.
4.3. second paragraph. This is way too long and doesn’t benefit from all that detail.

Minor comments

Throughout and most noticeably in lines 509 & 510 there is inconsistency with whether sentences are started with “A.” or “Amazona”

L41: change “relatively easy to” to “relative ease of”

L62: delete “of the Amazona” (redundant)

L75: delete “species” (redundant)

L79: delete “of geographic range and population size” (redundant)

L81: replace “the four Amazona” with “these four”

L92: change “directing” back to “direct”; change “strengthening” to “strengthen”

L364: delete “(in kilometres squared)” as this is in the table header

L386-8: delete “Panels show ... mean ?;” as this merely repeats the obvious in the figure.

L412: replace “Atlantic Forest Amazona population abundance” with “the abundance of Atlantic Forest Amazona species”

L427: insert “(Figure 3C)” after “counts”

L456,7: capitalise “parrot” for consistency

L478: replace “A. brasiliensis, of the smallest range,” with “With the smallest range, A. brasiliensis

L487-9: replace “species. Such ... or A. rhodocorytha—is” with “species—A. brasiliensis, A. pretrei or A. rhodocorytha. This does not mean they are”

L489: delete “For one,” (there is no no. 2)

L494: replace “they” with “forests”; delete “other”

L497-8: delete “Since among ... size,”

L498: replace “observed” with “report”

L498: delete “potentially” (tautology of “could”)

Table S1: why are the references bracketed?

Author Response

Response to Reviewer’s comments on Manuscript ID: diversity-1267950

Comments are marked as ‘REV1’ as they were written by Reviewer #1. Our responses to comments are marked ‘R’ and numbered.

Reviewer 1:

 REV1: Your revision is a considerable improvement, and I’ve mostly only minor points of expression to suggest. However, I feel that your Discussion still leaves a deal to be desired.

Sub-headings in the Discussion certainly help, though these need some re-organisation and amendment. Your use of “Temporal” in sub-headings 4.3 and 4.4 is non-specific and confounds two very different issues. Also, the material in section 4.3 conflates seasonal change with long-term change. I recommend:

     4.3. Seasonal change

     4.4. Long-term trends

and moving relevant material from 4.3 to 4.4.

You have not shortened your Discussion – actually lengthened it a little – and it is often not well focussed on your findings. In a number of paragraphs there’s a confusing multiplicity of themes resulting in them being too long. For each paragraph, I urge you to identify its theme in a crisp first sentence (be quite specific to something arising directly from your results), then elaborate that theme and delete ALL extraneous material (if important, move the extraneous material to a new paragraph). Avoid sweeping generalities in first sentences – some might be moved to the end of paragraph to form a conclusion to it, but beware overstating your findings.

R1: We added two new headings and revised the whole discussion. It is now 469 words shorter than in the previous version.

REV1: Throughout: I urge you to avoid incidentals and attempting to review every reference, instead sticking to key generalities relating directly to your results. If you feel all that detail is important to put on record, move it to Supplementary materials.

R2: Thank you. We took this advice into account when shortening the discussion.

Here’s some specific issues:

REV1: - 4.1. second paragraph. So what about sympatry or lack thereof? (And it isn’t in your Results.) Delete first section of this paragraph and identify the paragraph theme (range relationships to forest cover) in the first sentence.

R3: We deleted the comments about sympatry and focused the paragraph on range relationships to environmental factors.

REV1: - 4.2. first paragraph (missing lines?): this paragraph has two distinct themes which are confusingly interwoven: actual population counts and estimates, and the problem with variation in survey effort. I recommend two paragraphs, the first dealing with counts and estimates, i.e. sentences on lines 499-503 and 509-528, the second cautioning with issues arising from variation in survey effort.

R4: We cut 200 words off section 4.2 and reorganized the information in two paragraphs, one focusing on A. vinacea and A. rhodocorytha, which have statistical estimates of population size, and the other on A. pretrei and A. brasiliensis, for whom population size considerations must be based solely on counts. We thank the reviewer for the attention to this section which indeed needed cleaning and shortening.

REV1: - 4.2. second paragraph (see also below). This makes much of little. Regardless of the generality outlined by Holt et al., I can think of so many exceptions that the point is trivial unless specifically framed in terms of contrasting ecologies – which is well beyond the scope of your paper. At the very least, this paragraph would benefit from considerable shortening. Further, the first sentence is confusing. Do you mean “Density estimates will vary with estimates of both population and geographic range.”? But actually I feel that the first two sentences could and should simply be deleted. Certainly, they don’t belong at the beginning of the paragraph because the paragraph topic is the “interesting pattern”.

R5: We got rid of this paragraph, reducing the idea to one short comment at the end of the paragraph about A. pretrei and A. brasiliensis, mentioned in R4.

REV1: 4.3. second paragraph. This is way too long and doesn’t benefit from all that detail.

R6: We cut out 100 words from this paragraph.

Minor comments

REV1: Throughout and most noticeably in lines 509 & 510 there is inconsistency with whether sentences are started with “A.” or “Amazona”

R7: Thank you! We noticed and corrected all the inconsistencies in the lines 111, 228, 231, 290, 295, 399, 590, 902, and 914.

REV1: L41: change “relatively easy to” to “relative ease of”

R8: Changed to “relative ease of” in the line 41.

REV1: L62: delete “of the Amazona” (redundant)

R9: Deleted “of the Amazona” in the line 47.

REV1: L75: delete “species” (redundant)

R10: Deleted in the line 60.

REV1: L79: delete “of geographic range and population size” (redundant)

R11: Deleted in the line 64.

REV1: L81: replace “the four Amazona” with “these four”

R12: Replaced in the line 65.

REV1: L92: change “directing” back to “direct”; change “strengthening” to “strengthen”

R13: Changed in the lines 76 and 77.

REV1: L364: delete “(in kilometres squared)” as this is in the table header

R14: Deleted in the line 251.

REV1: L386-8: delete “Panels show ... mean ?;” as this merely repeats the obvious in the figure.

R15: Deleted in the line 262.

REV1: L412: replace “Atlantic Forest Amazona population abundance” with “the abundance of Atlantic Forest Amazona species”

R16: Replaced in the lines 287-288.

REV1: L427: insert “(Figure 3C)” after “counts”

R17: Inserted in the line 302-303.

REV1: L456,7: capitalise “parrot” for consistency

R18: Changed in the lines 335-336.

REV1: L478: replace “A. brasiliensis, of the smallest range,” with “With the smallest range, A. brasiliensis”

R19: Sentence deleted to reduce the paragraph.

REV1: L487-9: replace “species. Such ... or A. rhodocorytha—is” with “species—A. brasiliensis, A. pretrei or A. rhodocorytha. This does not mean they are”

R20: Replaced in the lines 373-375.

REV1: L489: delete “For one,” (there is no no. 2)

R21: Deleted in the line 375.

REV1: L494: replace “they” with “forests”; delete “other”

R22: In this sentence, “they” is related with forest and altitude, not only forest. We decided to maintain the word “they” and excluded the word “other”.

REV1: L497-8: delete “Since among ... size,”

R23: We rewrite this sentence in the lines 382-383.

REV1: L498: replace “observed” with “report”

R24: We rewrite this section to reduce the paragraph and the word was excluded. 

REV1: L498: delete “potentially” (tautology of “could”)

R25: We rewrite this section to reduce the paragraph and the word was excluded. 

REV1: Table S1: why are the references bracketed?

R26: We deleted the brackets in the references.

Reviewer 2 Report

I thank the authors for carefully addressing my earlier comments. I only suggest they include their justification for studying these counts (e.g., "Counts offer reasonable lower bound for population size and the variation in the length of the time series gives an illustration of what species have received more or less attention from the research community") in the methods and discussion.

Author Response

Response to Reviewer’s comments on Manuscript ID: diversity-1267950

Comments are marked as ‘REV2’ as they were written by Reviewer #2. Our responses to comments are marked ‘R’ and numbered.

Reviewer 2:

 REV2: I thank the authors for carefully addressing my earlier comments. I only suggest they include their justification for studying these counts (e.g., "Counts offer reasonable lower bound for population size and the variation in the length of the time series gives an illustration of what species have received more or less attention from the research community") in the methods and discussion.

R1: We thank the reviewer for the comments. We added the suggested sentence in the lines 398-399 of the Discussion, which now writes: “In the absence of statistical estimates, however, counts offer a reasonable lower bound for population size.”

Reviewer 3 Report

The revised article is clear, and the authors were included all the changes and given response for all the comments. Since, few clarification and corrections are necessary to improve the article.

Minor comments

  1. The authors should verify the genus name of the species, which is indicated in the revised article. Some places were mentioned “Amazona” and A. pretrei. For example: Line 392 and 414 indicated differently. The authors have to follow the naming all over the manuscript, once the abbreviation was introduced at the beginning.
  2. Line 258: include the country name “Brazil” at the end of the Institute name, before the web link.
  3. Verify the word “Extent area” in the revised article. Is that extent area or Extant area? Because, the meaning of the word is different. The authors were using the extent word in the line 68, 83, 111, 506.

Major comments

  1. The authors should verify all the reference again. The chronological order is not well clear in the discussion. Several references are in the figure 3 legend, please cite in the main text. Because the ref.68 was cited in the beginning of the paragraph of discussion (4.1. geographical range), not informed ref.64, 65, 66, 67 after the result part.
  2. Similarly, the chronological order of reference 35 was missing in the result part. The authors have indicated until ref.34 in the methodology part and citing ref. 35 (figure 3 legend) after ref. 36, 37. Please rewrite and modify the reference for the clarity to the readers.
  3. The reference 64 is not be able to find out in main text of the article. Verify and cite it correctly.

Author Response

Response to Reviewer’s comments on Manuscript ID: diversity-1267950

Comments are marked as ‘REV3’ as they were written by Reviewer #3. Our responses to comments are marked ‘R’ and numbered.

Reviewer 3:

REV3: The revised article is clear, and the authors were included all the changes and given response for all the comments. Since, few clarification and corrections are necessary to improve the article.

R1: We thank the reviewer for the comments.

Minor comments

REV3: 1. The authors should verify the genus name of the species, which is indicated in the revised article. Some places were mentioned “Amazona” and A. pretrei. For example: Line 392 and 414 indicated differently. The authors have to follow the naming all over the manuscript, once the abbreviation was introduced at the beginning.

R2: We noticed and corrected all the inconsistencies in the lines 111, 228, 231, 290, 295, 399, 590, 902, and 914.

REV3: 2. Line 258: include the country name “Brazil” at the end of the Institute name, before the web link.

R3: We added the information in the sentence (lines 177-179), which now writes: “We obtained Atlantic Forest cover data from Ribeiro et al. (in prep.), and Dense Forest cover data from the brazilian Instituto Brasileiro de Geografia e Estatística (https://www.ibge.gov.br/).”

REV3: 3. Verify the word “Extent area” in the revised article. Is that extent area or Extant area? Because, the meaning of the word is different. The authors were using the extent word in the line 68, 83, 111, 506.

R4: We revised the use of the word extent in the lines 53, 68, 90, and 95. We are using in a correct way, because we are referring to the extension itself. The word Extant is used only when we are referring to the area delimited by the IUCN where the species is resident.

Major comments

REV3: 1. The authors should verify all the reference again. The chronological order is not well clear in the discussion. Several references are in the figure 3 legend, please cite in the main text. Because the ref.68 was cited in the beginning of the paragraph of discussion (4.1. geographical range), not informed ref.64, 65, 66, 67 after the result part.

R5: We added most of the references from the figure description into the main text in the lines 295-296, 299, and 302. We believe that they are cited in a correct order now.

REV3: 2. Similarly, the chronological order of reference 35 was missing in the result part. The authors have indicated until ref.34 in the methodology part and citing ref. 35 (figure 3 legend) after ref. 36, 37. Please rewrite and modify the reference for the clarity to the readers.

R6: The reference 35 appears the first time in the methods. We checked the chronological order and it is correct.

REV3: 3. The reference 64 is not be able to find out in main text of the article. Verify and cite it correctly.

R7: The reference 64 appears the first time in the figure description in the line 353 and it is cited correctly.

Round 3

Reviewer 1 Report

I have only a few minor suggestions:

l41: change “relatively” to “relative”

l178: capitalise “brazilian”

Table 1 is cut off

l521: insert comma after “availability”

l530: delete comma after “[39]”

l697: delete “up” (redundant to “increased”)

l722: replace “on” with “in”

Author Response

Response to Reviewer’s comments on Manuscript ID: diversity-1267950

Comments are marked as ‘REV1’ as they were written by Reviewer #1. Our responses to comments are marked ‘R’ and numbered.

Reviewer 1:

Comments and Suggestions for Authors

REV1: I have only a few minor suggestions:

R1: We thank the reviewer for the careful revision. 

REV1: l41: change “relatively” to “relative”

R2: Changed to “relative” in the line 41.

REV1: l178: capitalise “brazilian”

R3: Brazilian capitalized in the line 168.

REV1: Table 1 is cut off

R4: We notice that this problem occurred only in the PDF version. The table is correct in the Word version.

REV1: l521: insert comma after “availability”

R5: Comma inserted in the line 375.

REV1: l530: delete comma after “[39]”

R6: Comma deleted in the line 384.

REV1: l697: delete “up” (redundant to “increased”)

R7: Word “up” deleted in the line 429.

REV1: l722: replace “on” with “in”

R8: Word “on” replaced to “in” in the line 454.

Reviewer 3 Report

The authors were given responses for all questions raised by the reviewer. I would like to appreciate the authors for accepting and improving the article.

Author Response

We thank the reviewer for the comments and suggestions.